# Quantifying In-Host Quasispecies Evolution

**DOI:** 10.3390/ijms24021301

**Published:** 2023-01-09

**Authors:** Josep Gregori, Marta Ibañez-Lligoña, Josep Quer

**Affiliations:** 1Liver Diseases-Viral Hepatitis, Liver Unit, Vall d’Hebron Institut de Recerca (VHIR), Vall d’Hebron Barcelona Hospital Campus, Passeig Vall d’Hebron 119-129, 08035 Barcelona, Spain; 2Centro de Investigación Biomédica en Red de Enfermedades Hepáticas y Digestivas (CIBERehd), Instituto de Salud Carlos III, Av. Monforte de Lemos, 3-5, 28029 Madrid, Spain; 3Biochemistry and Molecular Biology Department, Universitat Autònoma de Barcelona (UAB), Campus de la UAB, Plaça Cívica, 08193 Bellaterra, Spain

**Keywords:** quasispecies evolution, distributions similarity, quasispecies fitness partition, viral treatment, mutagenesis

## Abstract

What takes decades, centuries or millennia to happen with a natural ecosystem, it takes only days, weeks or months with a replicating viral quasispecies in a host, especially when under treatment. Some methods to quantify the evolution of a quasispecies are introduced and discussed, along with simple simulated examples to help in the interpretation and understanding of the results. The proposed methods treat the molecules in a quasispecies as individuals of competing species in an ecosystem, where the haplotypes are the competing species, and the ecosystem is the quasispecies in a host, and the evolution of the system is quantified by monitoring changes in haplotype frequencies. The correlation between the proposed indices is also discussed, and the R code used to generate the simulations, the data and the plots is provided. The virtues of the proposed indices are finally shown on a clinical case.

## 1. Introduction

All viruses that pass through an RNA replication phase are found in what is known as a quasispecies. That is, a set of closely related genomes that may exhibit a huge number of variants but keeping a high degree of similarity among them in a host. These variants are produced during the replication by the RNA-dependent RNA polymerases, which are error prone and lack the mechanism of error correction typical in most DNA polymerases [1].

Quasispecies are dynamical entities subject to evolution, generating new variants at each replication cycle, while losing the less fit and those unable to replicate. A quasispecies at a given time point may be described in molecular terms by the existing different genomes (haplotypes) and their frequencies (the number or fraction of molecules with the same sequence), the haplotype distribution. That is, a multinomial distribution where each category corresponds to a different haplotype. The evolution of this dynamic entity may then be represented by the changes observed in this distribution, as new categories appear and others disappear, and as their frequencies vary.

The extent of changes of a quasispecies in a host, between two time points, may be quantified by the genetic distance between the two viral populations [2], by the changes in quasispecies diversity indices [3], and by the distance or dissimilarity between the two haplotype distributions [4]. In this report, we discuss three selected indices used to compute the similarity between two haplotype distributions and their implications. With quasispecies simulated data, we show their particularities and correlations, and use plots to help in the interpretation of results. Finally a clinical HEV dataset, from a recent publication, is used to illustrate the practical use of these indices. They are particularly useful in the clinical follow up of a patient, where the compared quasispecies are highly related, and where the genetic distance between them may not suffice to describe the observed changes.

In the context of NGS, we denote each distinct genome as an *haplotype*, and each molecule sampled as a *read*. We shall be using this terminology throughout the paper.

## 2. Results

### 2.1. Simulated Pairs of Quasispecies

To quantify the extend of changes in a quasispecies, we compare the quasispecies composition at two time points. The pairs of quasispecies used to illustrate the results and discussion are obtained by a simple simulation with a limited number of haplotypes whose frequencies vary randomly within given constraints, and where a random number of these haplotypes are common to both quasispecies. The simulation aims to obtain closely related quasispecies as we could find, a few weeks or months apart, in a host. We simulate 10,000 pairs of related quasispecies, computing their similarity, and genetic distance. The simulated pairs are illustrated in the form of a table and a figure, confronting the haplotype distributions in both quasispecies, as in Table 1 and Figure 1.

The index of Commons, $C_{m}$ (Equation (Equation 1)), is strongly indicative of quasispecies relatedness. When the two quasispecies have all their haplotypes identical, this index results in a value of 1, even when the proportions are highly dissimilar. On the other hand the Overlap index, $O_{v}$ (Equation (Equation 2)), may result in low values even when all haplotypes of both quasispecies are identical. Finally, the Yue–Clayton index, $Y_{C}$ (Equation (Equation 3)), results in high values when the fraction of common haplotypes is high, and their proportions are similar. The overlap between distributions is better illustrated with a plot like Appendix A.

A summary of the values of similarity indices obtained from the simulated quasispecies pairs is given in Table 2, along with the number of pairs resulting in a similarity value over 0.5, 0.75 and 0.9. The histograms for the three indices are given in Appendix A. On the other hand, Table 3 and Figure 2 show the distribution of the three indices for the 2698 simulated pairs resulting in $C_{m}$ values over 0.75, that is, highly related. The histograms for the corresponding nucleotide diversities and genetic distances (Equations (Equation 4)–(Equation 7)) are given in Appendix A.

#### 2.1.1. Correlations

The eventual redundancy in the information provided by these indices, and by the genetic distance, may be assessed by inspecting the correlation coefficient between them, taking the 10,000 simulated values, as in Table 4.

These correlations may be further illustrated by the joint density plots in Figure 3, Figure 4, Figure 5 and Figure 6, from which result the following observations:$C_{m}$ and $O_{v}$: At high $O_{v}$ values only high values of $C_{m}$ may occur, on the other hand at very low $O_{v}$ values almost all $C_{m}$ values are possible. This is consistent with the definition of both indices.$Y_{C}$ and $O_{v}$: are highly correlated and seemingly do not convey significant additional information.$C_{m}$ and $D_{A}$: At low $C_{m}$ values, $D_{A}$ takes high values, but at high $C_{m}$ values, $D_{A}$ spans the highest range of values. The lower $D_{A}$ values correspond to high values of $C_{m}$.$O_{v}$ and $D_{A}$: At high $O_{v}$ values, $D_{A}$ takes the lower values, but at low $O_{v}$ values, $D_{A}$ spans a high range of values.

The information provided by $C_{m}$, $O_{v}$, and $D_{A}$ complement each other and offer different faces of the same comparison. $C_{m}$ expresses how related the two quasispecies are, in the sense of having common haplotypes, even if the frequencies are different. $O_{v}$ expresses how similar both distributions are, both in haplotypes and frequencies. Additionally, $D_{A}$ provides the net genetic distance between the two quasispecies, taken as populations of viruses.

#### 2.1.2. Illustration of Selected Pairs

From the set of simulated pairs, a few are selected attending to the values of the three indices, to help in the understanding and interpretation of these indices, and are plotted in Appendix A. Table 5 shows a summary of these examples.

To improve the visualization of haplotypes unique to either quasispecies, the proportions of both quasispecies are sorted according to the order in decreasing value of the proportions of the quasispecies A. The haplotypes unique to quasispecies B will be placed on the right of the plot, or the bottom of the table.

These selected pairs are shown on the joint density plots of $C_{m}$ and $D_{A}$ in Appendix A, and $C_{m}$ and $O_{v}$ in Appendix A, in order to illustrate their position with respect to the bulk of the simulation.

Appendix A show two typical examples with high values in the three indices.

Appendix A show two examples with high $C_{m}$ but very low $O_{v}$ and $Y_{C}$ values. This situation arises when there is a number of common haplotypes with mid to high proportions in one quasispecies and very low in the second.

Appendix A show intermediate cases with $C_{m}$ below 0.70, and feeble values of $O_{v}$ and $Y_{C}$.

Finally, Appendix A show cases with higher values of $O_{v}$ and $Y_{C}$.

### 2.2. Simulated Treatment

As described under methods, the evolution of a quasispecies with a shrinking dominant haplotype, an emerging haplotype, and a set of minority haplotypes subject to quasispecies dynamics was simulated. The result is represented in Figure 7, where the evolution in the frequencies of each of the 40 haplotypes is shown at each of the simulated evolution steps, with the corresponding dominant haplotype labelled with a + sign.

The fitness partition (QFP) analysis applied to the simulated samples in the follow-up example (Figure 8) show the four fitness fractions (QFF) in the form of a shrinking dominant haplotype in parallel with an increasing volume of molecules belonging to emergent haplotypes as a side effect of a treatment, generating resistant variants. This figure constitutes a summary of the full quasispecies distributions illustrated in Figure 7.

The similarity indices discussed above ($C_{m}$ Equation (Equation 1), $O_{v}$ Equation (Equation 2), and $Y_{C}$ Equation (Equation 3)), take values from 0 to 1, and may be easily transformed into distances by the rule Distance=1−Similarity. Figure 9 illustrates the matrix of Yue-Clayton distances for the simulated treatment.

These distances may then be used to construct quasispecies dendrograms, or transformed by multidimensional scaling (MDS) to plot maps showing the relationships between the quasispecies. $D_{A}$ genetic distances (Equation (Equation 7)) may be used in the same way to get dendrograms or MDS maps, as shown in Appendix A.

Note that by the very definition of the simulation used in the follow-up example, all quasispecies pairs show a $C_{m}$ similarity index of 1, as all samples in the series share the same 40 haplotypes, although at varying frequencies. In real cases both the $C_{m}$, the $O_{v}$ or the $Y_{c}$, and the $D_{A}$ will be informative about the quasispecies evolution, showing different aspects about the changes produced. Additionally, the QFF contributes an interesting summary of quasispecies evolution.

### 2.3. A Clinical Case

This is the clinical follow-up of a patient chronically infected by HEV who underwent an off-label treatment with Ribavirin for three years [5]. The treatment involved three regimens (600, 800, and 1000 mg/day) with discontinuations caused by adverse effects, followed by relapses.

This dataset is of particular interest here, because it involves the follow-up of a patient infected by a zoonotic virus, HEV, treated with a mutagenic agent, with the follow-up spanning over three years of treatment. In this case, the naturally high genetic diversity of HEV quasispecies is enhanced by the treatment with a mutagen.

The behavior of the three indices, in this case, is illustrated in Figure 10, where the similarities between each pair of sequential samples is shown, comparing sequential haplotype distributions on the left, and corresponding phenotype distributions on the right. The impact of the mutagenic treatment is evidenced by the sequential decrease in $C_{m}$, whose behavior is smoother than that of $O_{v}$ or $Y_{C}$. The continued decrease in $C_{m}$ value indicates that the proportion of molecules with sequences corresponding to haplotypes common to the two compared quasispecies is shrinking, consistent with the expected results of a mutagenic treatment, which generates new variants at an enhanced rate. The new variants will increase in abundance or fade according to their replicative fitness. The drop in $C_{m}$ is especially marked when each treatment is initiated, especially those at 800 and 1000 mg/day, but these are followed by a small correction upwards. On the other hand, despite the radical changes observed in the haplotype composition, the analysis by phenotype composition shows that the functionality was maintained over a significant period of time, thanks to the generation of a rich set synonymous variants, and until the 800 mg/day regimen took effect. The changes observed in phenotype composition near the end of treatment, together with the observed increase in viral load may indicate that, either some resistance to the treatment was generated, or that the rich set of synonymous haplotypes generated and selected during the treatments contributed to generate a more resilient quasispecies [6]. The similarity in the phenotype distribution between the end-of-treatment sample and that taken one year after is very high. This figure also shows that the indices $O_{v}$ and $Y_{C}$ are highly correlated, as previously shown with the simulated data. The $C_{m}$ and $O_{v}$ similarities in this dataset are plotted in Figure 11 over the 2D-density of the simulated data to show the correspondence between this clinical case and the simulated data.

The QFF profile of haplotypes and phenotypes of this case was presented and analyzed in the previous publication [5], and provides an interesting complementary and consistent view of this quasispecies evolution.

## 3. Discussion

The proposed methods are intended to be used in the analysis of changes occurred in a in-host quasispecies along time, as a consequence of the host immune system or of an external action, like a treatment. The quasispecies are treated as entities (closed ecosystems or genetic populations), where the respective distribution of molecules are compared, in contrast with the more widespread comparison of summary values such as diversity indices (i.e., Shannon entropy), or of genetic diversification (i.e., nucleotide diversity) [3].

In a recent paper [5], we introduced the Quasispecies Fitness Partition (QFP) in four fractions (QFF), also described under methods, and we recommended its use together with the Hill Numbers Profile (HNP) to visualize the evolution of a quasispecies. Those methods were used in a deep exploration of a clinical case of an HEV infection treated with ribavirin. As part of the discussion, we proposed the use of distances between haplotype distributions as an alternative or complement to the use of genetic distances between quasispecies. This paper comes to explore three selected indices of similarity between haplotype distributions, from which the corresponding distances may be obtained.

Here, we have used simulated data aimed only at producing closely related quasispecies, similar to what could be observed in the follow-up of a single patient, with enough simplicity to be tabulated and plotted. However, to put in clinical context the methods here described, we have added the data of a clinical follow-up of an HEV chronically infected patient treated with a mutagen, spanning three years of observation, and different treatment regimens. Since HEV is an RNA virus having very high mutation rates, on the range of $10^{-3}$ to $10^{-4}$ substitutions/base/replication cycle [7], similar to other highly clinical relevant viruses such as HCV or HIV, the tools presented can be extrapolated to the vast majority if not all RNA viral infections.

The simulation of a substantial number of paired quasispecies allowed us to illustrate particular cases of interest, contributing to the interpretation of results, and also to estimate the correlations between the three indices ($C_{M}$, $O_{v}$ and $Y_{C}$), and with the quasispecies genetic distance, $D_{A}$. The correlation values show the pairs $O_{v}$ and $Y_{C}$, $O_{v}$ and $D_{A}$, and $Y_{C}$ and $D_{A}$ as highly correlated, with $C_{m}$ the most independent of the others. Despite this high correlation we recommend the use of three distances, $C_{m}$, $O_{v}$ or $Y_{C}$, and $D_{A}$. Nevertheless for distant quasispecies the four distances will contribute valuable information.

The use of these distances is shown with the simulated data of a quasispecies treatment (Figure 7), the changes experienced by the quasispecies with samples taken at given evolutionary steps are summarized in the QFF plot, Figure 8. The relationship between the quasispecies is shown in the form of a matrix of $Y_{C}$ distances, Figure 9, from which we obtain a dendrogram by hierarchical clustering with the average method, Appendix A, or a MDS map, Appendix A. Using $D_{A}$ distances we may obtain an alternative dendrogram, Appendix A, or an alternative MDS plot, Appendix A.

A key point with all these methods is the availability of quasispecies haplotypes with corresponding frequencies. The classical and more widespread NGS data analysis procedures for viruses, like Galaxy [8], i.e., limit sequencing errors by trimming the reads at their ends, where the quality is poorer, by a number of nucleotides, attending to instrument quality scores, using different algorithms. As a result of this trimming the coverages are uneven, even within the same amplicon, which prevents the direct obtention of amplicon haplotypes. In [5,9], for instance, we describe the method used by our group to obtain high quality amplicon haplotypes in sequencing viral quasispecies samples. It is simply based on respecting the integrity of full reads, with no trimming, except for the primers. The quality filters are executed on full reads. This requires high sequencing quality and very high coverage to get a comprehensive picture of an infection that may involve viral loads higher than $10^{6}$ copies/mL of blood. Currently we are only able to obtain high quality amplicon haplotypes of a size slightly over 500 bp, with coverages of the order of $10^{5}$ reads per amplicon, sequencing with Illumina instruments. Despite this limitation, quasispecies genomes may be studied amplicon by amplicon. On the other hand, when the monitored treatment is by a direct acting agent that targets a specific region of the genome a single amplicon may suffice [9]. There are a number of inferential methods for reconstructing full viral haplotypes from short reads, but they have limitations, require of special computational resources for high coverages, and perform poorly with samples of high genetic diversity, according to a recent review evaluation of them [10].

The clinical case presented has given the opportunity to show a practical application of the proposed methods. This dataset with thousands of haplotypes in each sample, and coverages in the range of 5 $\times \;10^{4}$ to 5 $\times \;10^{5}$ reads, shows a correlation between the three indices consistent with what has been observed with the more modest simulated pairs of quasispecies entailing very few haplotypes; nevertheless, a critical aspect in the simulations was to ensure a close relationship between pairs of quasispecies, as it is the case in the follow-up of a patient, the main objective of this work.

The advantage of the described methods is that they provide rich summaries and visual tools to monitor the changes occurring in a viral quasispecies at the molecular level, with time. This facilitates the interpretation of the biological changes in the quasispecies, and also provides a means to diagnose possible outcomes of a treatment when monitoring a patient, as seen with the discussed HEV clinical case.

In the case of mutagenic treatments, we recommend this method, combined with the method of quasispecies fitness fractions (QFF), and the Hill numbers profile (HNP) [5]. When the quasispecies evolution rate is low compared to mutagenic scenarios, the QFF may result as insufficient to evidence changes in the quasispecies, and the proposed indices could be more sensitive to changes.

## 4. Materials and Methods

### 4.1. Data

#### 4.1.1. Simulation of Paired Quasispecies

To quantify the extent of changes (evolution) of a quasispecies, we compare the quasispecies composition at two time points. The paired quasispecies needed to illustrate the results and discussion are obtained by simulation as described in the following method:**Distribution pattern:** 20,000 random occurrences of a geometric distribution, with parameter p=0.2, are generated, simulating 20,000 reads of over 35 haplotypes. The frequencies of this distribution are used as pattern distribution on which to apply random selection criteria of frequencies.**Select frequencies for quasispecies A:** From the above pattern distribution, 12 frequencies are randomly selected to represent the composition of quasispecies A.**Select frequencies for quasispecies B:** From a new pattern distribution generated with the same parameters as above, randomly select 12 frequencies to represent the composition of quasispecies B.**Confront both simulated quasispecies:** The two quasispecies are composed together of 20 haplotypes, some common to both quasispecies, some unique to either one. Assign randomly the 12 frequencies of quasispecies A among the 20, and do the same with the 12 frequencies of quasispecies B. Remove from the 20 any haplotype not populated (0 reads in both quasispecies).

A single cycle of this simulation results in the distributions of two paired quasispecies, which are given as shown in Table 1, and may be represented, confronting both distributions, as in Figure 1. The chosen numbers of reads and haplotypes in the simulation are arbitrary, a simplification of real life cases, but complex enough to compose a quasispecies.

The simulated pairs of quasispecies are related because of the result of a random selection of 12 haplotypes each from a common source of 20. On the other hand, the random selection of frequencies results in varying proportions for each haplotype and varying coverages (total number of reads) for each quasispecies. In this way, in each pair, we consider quasispecies B as the result of an evolution from quasispecies A.

The R code is provided in the Appendix A.

#### 4.1.2. Simulation of a Viral Treatment Follow-Up

The previous simulation aimed to generate pairs of quasispecies, more or less distant, as a result of certain evolution from the first to the second, and it was intended to help in the understanding and interpretation of the similarity indices and the correlations between them.

A second simulated dataset aims to generate a sequence of quasispecies that could be the result of an external treatment which generates resistant variants as a side effect. The quasispecies will consist of 40 haplotypes of three types:The dominant haplotype, initially at a frequency of 99.9% evolving at a pace of a constant uniformly distributed between 0.85 and 1.05, at each evolution step.A minoritary haplotype initially at (0.1/39)%, and evolving at a pace of a constant uniformly distributed between 0.95 and 1.25, at each evolution step.The remaining 38 haplotypes, initially at (0.1/39)%, and evolving at a pace of a constant uniformly distribution between 0.8 and 2.5. Only a random number of these, between 2 and 10, are submitted to evolution at each step. The remaining are left as they were.

In this way, samples are sequentially generated at evolution steps 10, 20, 30, 35, 40, 45, 50, 55, 60, 65, 70, 75, and 80. The resulting haplotype distributions are plotted in Figure 7.

The R code is provided in the Appendix A.

#### 4.1.3. A Clinical HEV Case

This dataset is taken from a recent publication [5], which shows the negative effects of early treatment discontinuation by a mutagenic agent of an HEV chronically infected patient. This dataset is used to show an example of application of the proposed method to a practical case. Briefly, this is the clinical follow-up case of a 27-year-old patient who acquired chronic HEV infection after undergoing two kidney transplantations. The patient received three different RBV regimens (600 mg/day, 800 mg/day, and 1000 mg/day) with discontinuations caused by adverse effects, followed by relapses.

A single amplicon covering genomic positions 6323 to 6734 on the HEV ORF2 region was sequenced, for each of 13 sequential samples taken from May 2018 to June 2021. The coverage range of the final dataset is 53,307–503,770 reads, with a median of 328,271 reads per sample/amplicon, covering the full amplicon, and enabling the obtainment of amplicon haplotypes and corresponding frequencies. The number of haplotypes per sample are in the range 1688–7881, with a median number of 5602.

### 4.2. Methods

#### 4.2.1. Similarity between Distributions

The similarity between two distributions may be quantified by a rich set of different indices [4]. In this report, we use three of them:Commons: As the fraction of reads belonging to haplotypes populated in both quasispecies.
(1)$$C_{m}\;=\;\frac{1}{2}\sum_{i}\left\{\left(p_{i}+q_{i}\right)\;I\left(p_{i}>0\wedge q_{i}>0\right)\right\}$$Overlap: As the sum of the minimum proportion of common haplotypes.
(2)$$O_{v}\;=\;\sum_{i}min\left(p_{i},q_{i}\right)$$Yue–Clayton: This index takes fuller account of all proportion information, considering the proportions of both common and unique haplotypes. [11]
(3)$$Y_{C}\;=\;\frac{\sum_{i}p_{i}\;q_{i}}{\sum_{i}p_{i}^{2}+\sum_{i}q_{i}^{2}-\sum_{i}p_{i}\;q_{i}}$$

The three indices vary from 0 (no similitude) to 1 (equal quasispecies). The disimilarity, or distance, between two distributions may be computed as 1 minus the similarity index.

#### 4.2.2. Genetic Distance between Quasispecies

The nucleotide distance between two quasispecies [2], *X* and *Y*, may be estimated by:(4)$$D_{XY}\;=\;\sum_{i\in X}\;\sum_{j\in Y}\;p_{i}\;d_{ij}\;q_{j}$$
where $p_{i}$ and $q_{j}$ are the proportion of the *i*-th haplotype in quasispecies *X*, and that of the *j*-th haplotype in quasispecies *Y*, and $d_{ij}$ is the genetic distance between both haplotypes. The sum extends over all haplotypes in both quasispecies. This distance is interpreted as the average number of nucleotide substitutions between the reads from quasispecies *X* and quasispecies *Y*.

Taking into account the nucleotide diversity of each quasispecies [2], that is the average number of nucleotide substitutions for a random pair of reads in the quasispecies, $D_{X}$ and $D_{Y}$, which may be estimated by:(5)$$D_{X}\;=\;\frac{N_{X}}{N_{X}-1}\sum_{i\in X}\;\sum_{j\in X}\;p_{i}\;d_{ij}\;p_{j}$$
(6)$$D_{Y}\;=\;\frac{N_{Y}}{N_{Y}-1}\sum_{i\in Y}\;\sum_{j\in Y}\;q_{i}\;d_{ij}\;q_{j}$$
where $N_{X}$ and $N_{Y}$ are the number of reads in each quasispecies, then the net nucleotide substitutions between the two quasispecies [2] is estimated by:(7)$$D_{A}\;=D_{XY}-\left(D_{X}+D_{Y}\right)/2$$

$D_{A}$ will be taken as the genetic distance between two quasispecies.

The quasispecies pairs are simulated in a way that all haplotypes are considered to have a single substitution with respect to the master haplotype in the first quasispecies. In this way, the matrix of distances between all pairs of haplotypes in both quasispecies has the form:(8)$$D:\left\{\begin{array}{c} d_{ij}=0,\forall \;i=j\\ d_{ij}=1,\forall \;i=1\;and\;j>1\\ d_{ij}=1,\forall \;j=1\;and\;i>1\\ d_{ij}=2\;otherwise \end{array}\right\}$$

#### 4.2.3. Quasispecies Fitness Partition (QFP)

A quasispecies, at a given time, understood as a viral population, is usually comprised of a predominant haplotype, a few low- to medium-frequency genomes, various rare haplotypes with very low fitness but still able to replicate at some level, and some defective genomes unable to replicate. This composition can be modeled using the set of frequencies of all haplotypes in the quasispecies as parameters of a multinomial distribution, $\Pi =\{p_{1},p_{2},...,p_{n}\}$ with $\sum_{i=1}^{n}p_{i}=1$. Where $p_{i}$ is the frequency in the quasispecies of the *i*-th haplotype. The parameters, $p_{i}$, are sorted in decreasing order without a loss of generality.

In this way, the quasispecies can be partitioned into fractions limited by frequency thresholds of interest [5], as in is Equation (Equation 9), where a partition into four fractions (QFF) is illustrated, and where, $p_{1}^{\prime }$, $p_{2}^{\prime }$, $p_{3}^{\prime }$ and $p_{4}^{\prime }$ represent the four fractions.
(9)$$\begin{array}{c} \Pi_{1}=\left\{p_{1},p_{2},...,p_{k}\right\},\;\;\forall p_{i}:p_{i}\geq p_{k}\\ \Pi_{2}=\left\{p_{k+1},p_{k+2},...,p_{l}\right\},\;\;\forall p_{i}:p_{l}\leq p_{i}<p_{k}\\ \Pi_{3}=\left\{p_{l+1},p_{l+2},...,p_{m}\right\},\;\;\forall p_{i}:p_{m}\leq p_{i}<p_{l}\\ \Pi_{4}=\left\{p_{m+1},p_{m+2},...,p_{n}\right\},\;\;\forall p_{i}:p_{n}\leq p_{i}<p_{m} \end{array}$$
(10)$$\begin{array}{c} p_{1}^{\prime }=\sum_{i=1}^{k}p_{i};\;p_{2}^{\prime }=\sum_{i=k+1}^{l}p_{i};\;p_{3}^{\prime }=\sum_{i=l+1}^{m}p_{i};\;p_{4}^{\prime }=\sum_{i=m+1}^{n}p_{i} \end{array}$$
(11)$$\Pi^{\prime }=\left\{p_{1}^{\prime },p_{2}^{\prime },p_{3}^{\prime },p_{4}^{\prime }\right\}$$

In the typical quasispecies structure mentioned above, the four fractions can be defined as follows:Master: the fraction of molecules belonging to the most frequent haplotype; that is, the one present at the highest percentage ($p_{1}^{\prime }=p_{1}$).Emerging: the fraction of molecules present at a frequency greater then 1% and less than the master percentage, belonging to haplotypes that are potentially able to compete with the predominant one and possibly replace it ($p_{2}^{\prime }$).Low fitness: the fraction of molecules present at frequencies from 0.1% to 1%, belonging to haplotypes that have a low probability of progressing to higher frequencies ($p_{3}^{\prime }$).Very low fitness: the fraction of molecules present at frequencies below 0.1%, belonging to haplotypes with very low fitness and to defective genomes. The likely fate of these molecules individually is degradation, but the fraction is continuously fed with new very low fitness genomes produced by replication errors or by host editing activities ($p_{4}^{\prime }$).

This partition represents a summarization of the full haplotype distribution, where changes in each fraction have a straightforward biological meaning, and allow for the interpretation of the effects caused by the current environment, or by the administration of an external agent.

### 4.3. Software and Statistics

All computations were done in R (v4.0.3) [12], using packages ape [13], tidyverse [14], and ggplot2 [15]. The full code of the simulations and computations is provided in the Appendix A. The session info follows:


sessionInfo()



R version 4.0.3 (2020-10-10)



Platform: x86_64-w64-mingw32/x64 (64-bit)



Running under: Windows 10 x64 (build 19043)



Matrix products: default



Random number generation:



 RNG:     Mersenne-Twister



 Normal:  Inversion



 Sample:  Rounding



locale:



[1] LC_COLLATE=Catalan_Spain.1252  LC_CTYPE=Catalan_Spain.1252



[3] LC_MONETARY=Catalan_Spain.1252 LC_NUMERIC=C



[5] LC_TIME=Catalan_Spain.1252



attached base packages:



[1] stats     graphics  grDevices utils     datasets  methods   base



other attached packages:



[1] forcats_0.5.1   stringr_1.4.0   dplyr_1.0.7     purrr_0.3.4



[5] readr_2.0.0     tidyr_1.1.3     tibble_3.1.3    ggplot2_3.3.5



[9] tidyverse_1.3.1



loaded via a namespace (and not attached):



 [1] Rcpp_1.0.7       cellranger_1.1.0 pillar_1.6.2     compiler_4.0.2



 [5] dbplyr_2.1.1     tools_4.0.2      digest_0.6.27    jsonlite_1.7.2



 [9] lubridate_1.7.10 lifecycle_1.0.0  gtable_0.3.0     pkgconfig_2.0.3



[13] rlang_0.4.11     reprex_2.0.1     cli_3.0.1        rstudioapi_0.13



[17] DBI_1.1.1        haven_2.4.3      xml2_1.3.2       withr_2.4.2



[21] httr_1.4.2       fs_1.5.0         generics_0.1.0   vctrs_0.3.8



[25] hms_1.1.0        grid_4.0.2       tidyselect_1.1.1 glue_1.4.2



[29] R6_2.5.0         fansi_0.5.0      readxl_1.3.1     farver_2.1.0



[33] tzdb_0.1.2       modelr_0.1.8     magrittr_2.0.1   backports_1.2.1



[37] scales_1.1.1     ellipsis_0.3.2   rvest_1.0.1      assertthat_0.2.1



[41] colorspace_2.0-2 labeling_0.4.2   utf8_1.2.2       stringi_1.7.3



[45] munsell_0.5.0    broom_0.7.9      crayon_1.4.1


## Figures and Tables

**Figure 1 ijms-24-01301-f001:**
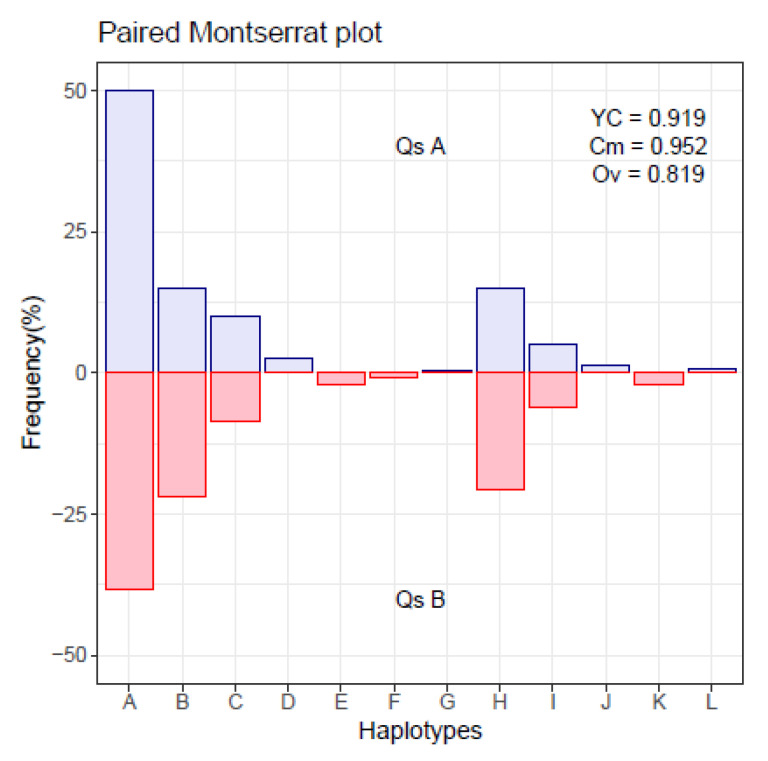
Montserrat plot with paired haplotype distribution.

**Figure 2 ijms-24-01301-f002:**
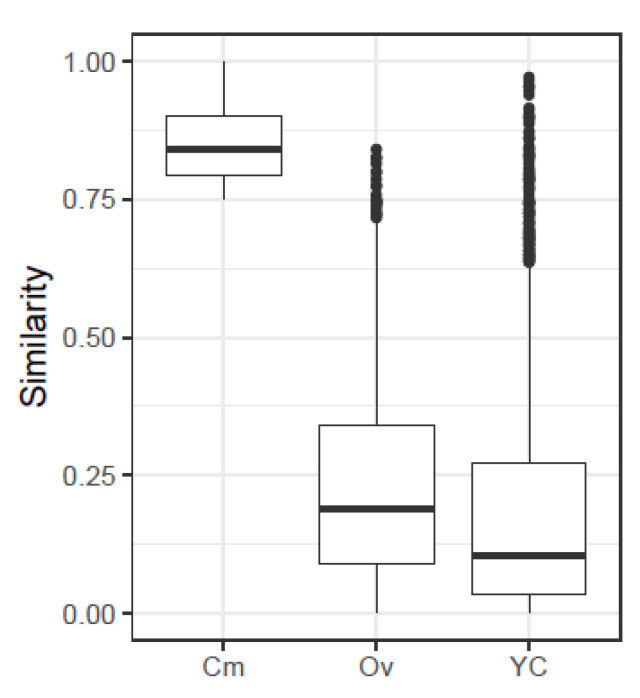
Boxplots with the distributions of the three indices, for all simulated pairs which result in values of $C_{m}$ over 0.75.

**Figure 3 ijms-24-01301-f003:**
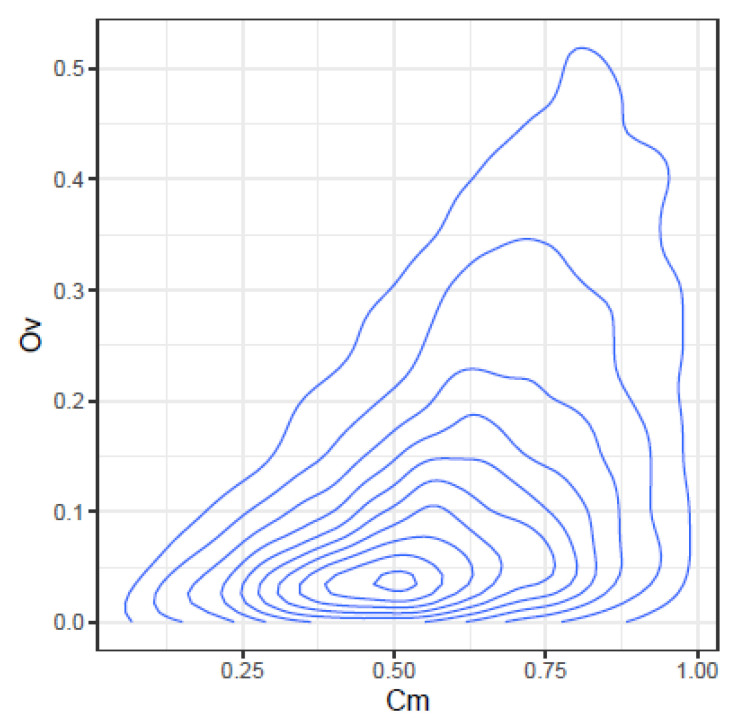
Joint density of $C_{m}$ and $O_{v}$.

**Figure 4 ijms-24-01301-f004:**
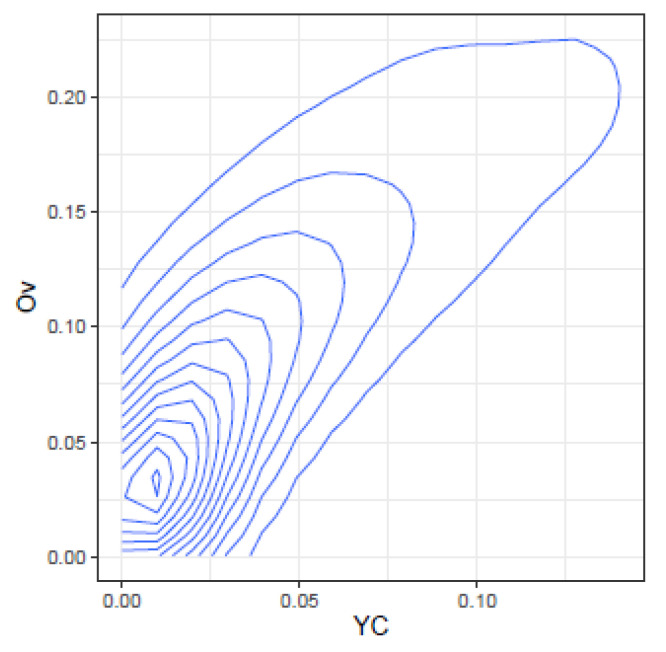
Joint density of $Y_{C}$ and $O_{v}$.

**Figure 5 ijms-24-01301-f005:**
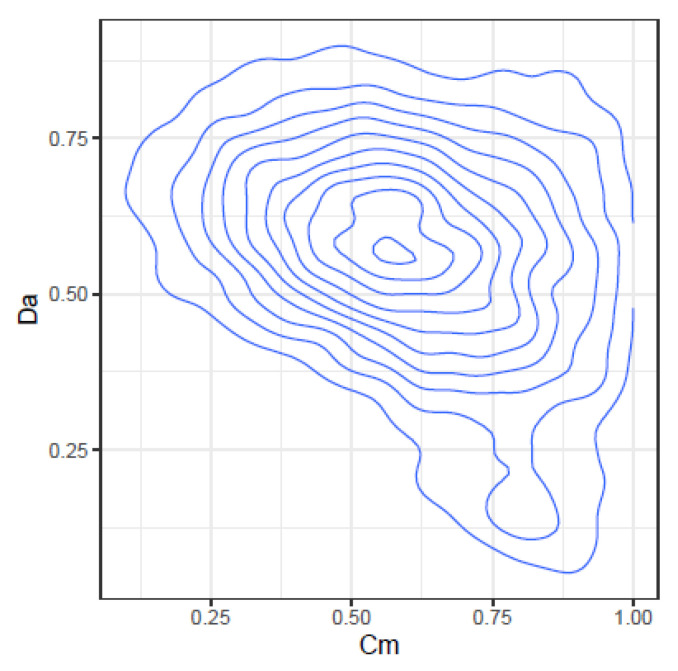
Joint density of $C_{m}$ and $D_{A}$.

**Figure 6 ijms-24-01301-f006:**
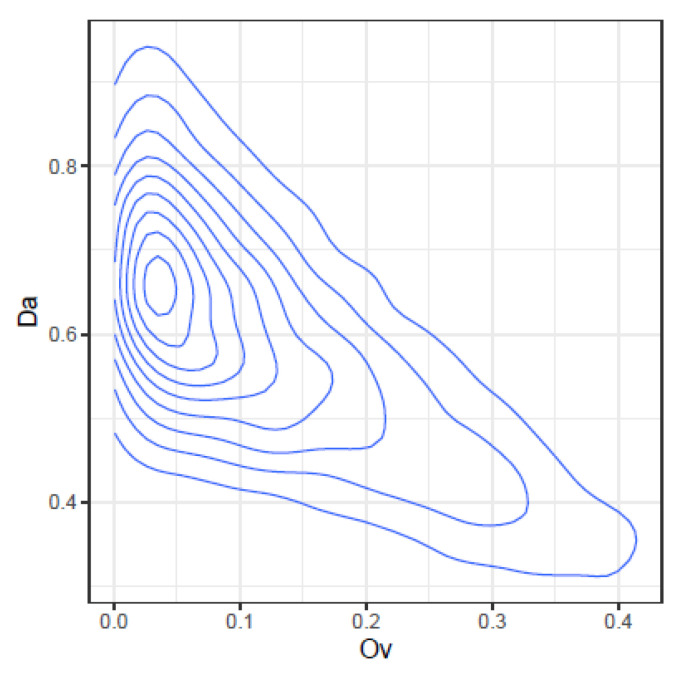
Joint density of $O_{v}$ and $D_{A}$.

**Figure 7 ijms-24-01301-f007:**
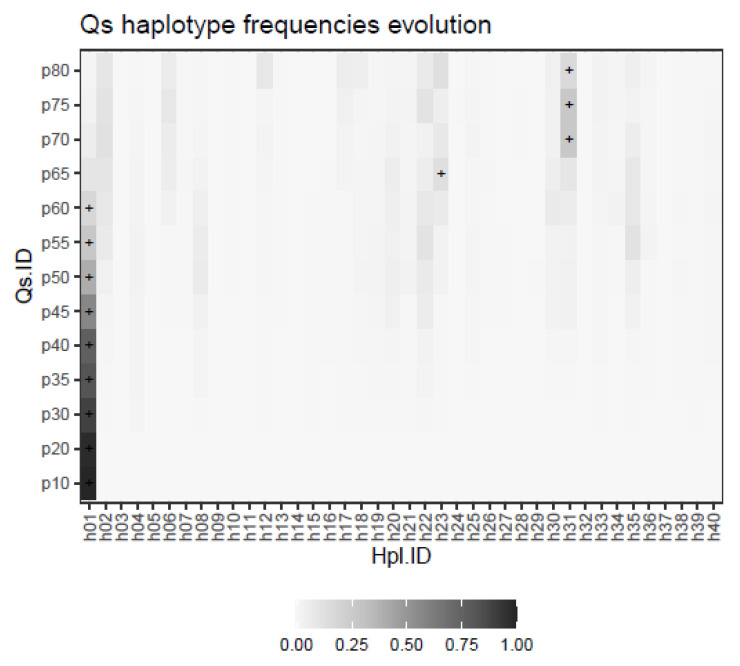
Haplotype distributions in the simulated follow-up example. The dominant haplotype at each step is labeled with a + sign.

**Figure 8 ijms-24-01301-f008:**
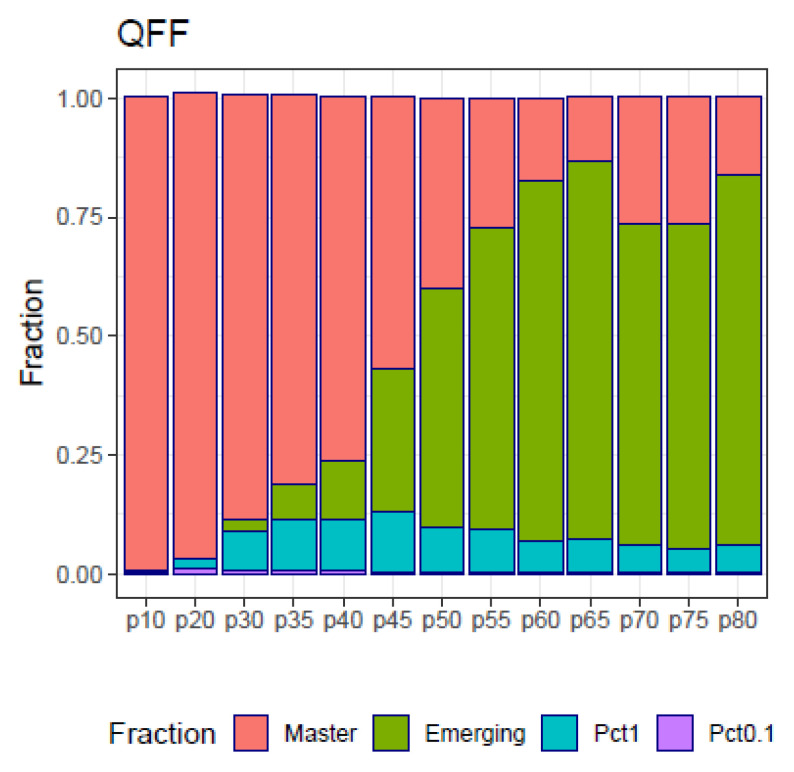
Quasispecies fitness partition of the simulated follow-up.

**Figure 9 ijms-24-01301-f009:**
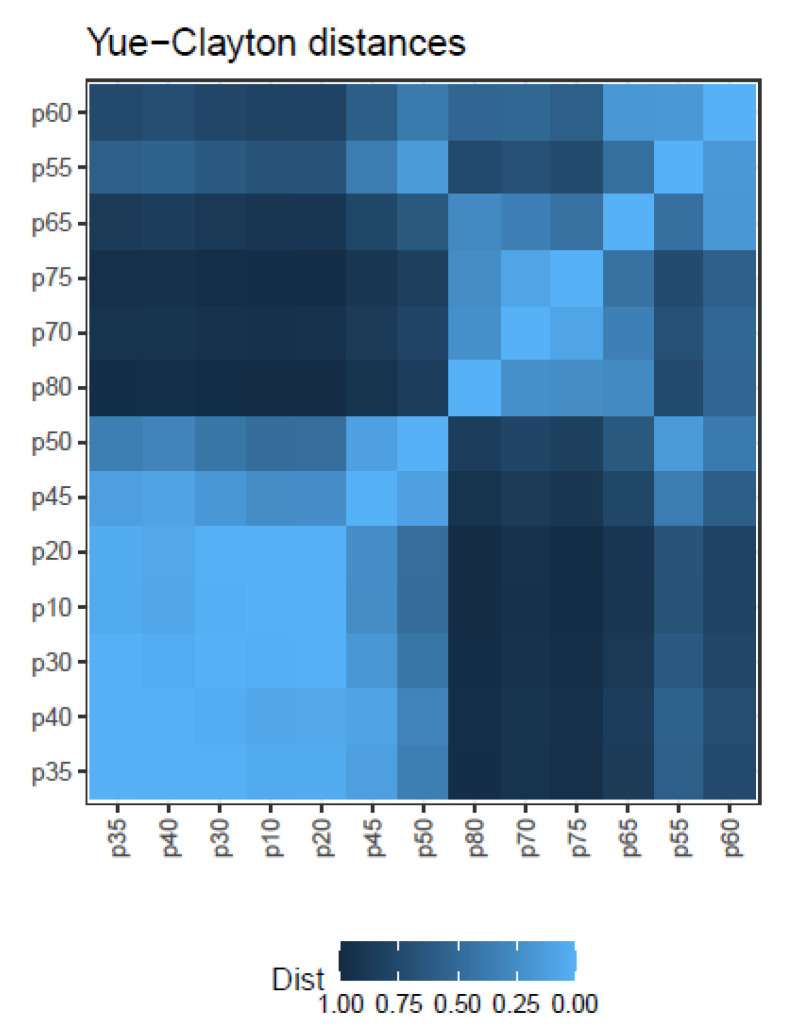
Matrix of Yue–Clayton distances between quasispecies haplotype distributions.

**Figure 10 ijms-24-01301-f010:**
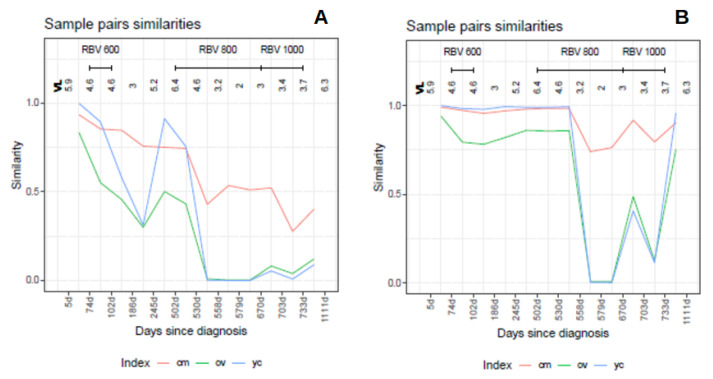
Distribution similarities in the composition of pairs of sequential samples in the HEV clinical case. (**A**) Haplotypes. (**B**) Phenotypes. Each point is the result of the comparison of two sequential samples, and is depicted in between the two compared samples. The segments above the figures show the time spanned by each treatment. Each sample is labeled as days since diagnosis. (VL viral load in logarithms, RBV ribavirin, cm $C_{m}$, ov $O_{v}$, yc $Y_{C}$).

**Figure 11 ijms-24-01301-f011:**
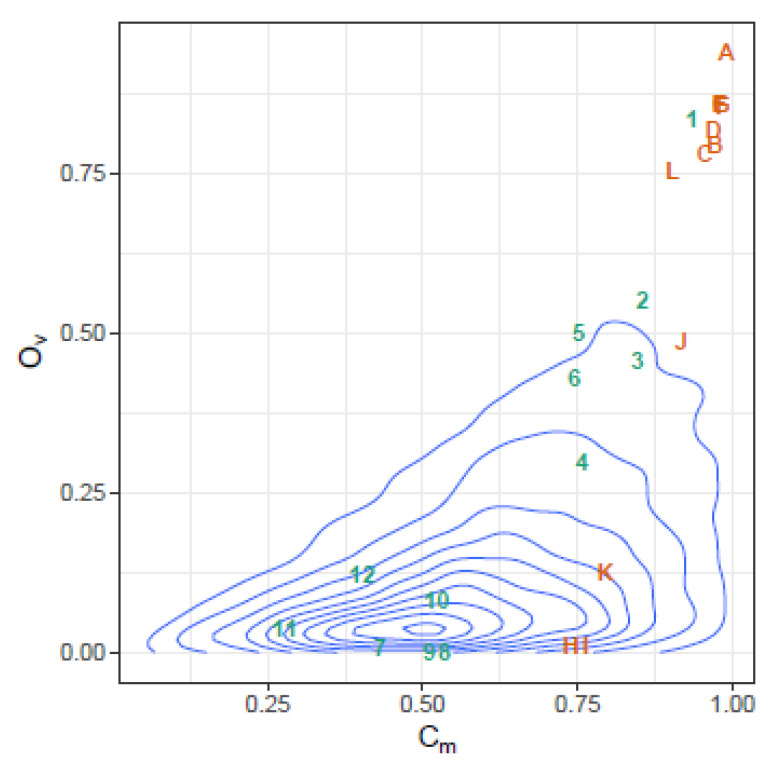
Observed similarities in the HEV clinical case plotted over the 2D-density of the simulated data. Similarities in haplotype distribution between pairs of sequential samples are labeled with numbers in increasing order, similarities in phenotype distribution are labeled with alphabetically ordered capital letters. The points on the top-right show high similarity by both indices.

**Table 1 ijms-24-01301-t001:** Two closely related quasispecies. Hpl haplotype ID, nA reads in quasispecies A, nB reads in quasispecies B, pA and pB corresponding frequencies (%).

Hpl	nA	nB	pA	pA
A	2000	1400	50.06	38.36
B	600	800	15.02	21.92
C	400	310	10.01	8.49
D	100	0	2.5	0
E	0	70	0	1.92
F	0	30	0	0.82
G	15	0	0.38	0
H	600	750	15.02	20.55
I	200	220	5.01	6.03
J	50	0	1.25	0
K	0	70	0	1.92
L	30	0	0.75	0

**Table 2 ijms-24-01301-t002:** Summary of similarity values between pairs of quasispecies.

	$O_{v}$	$C_{m}$	$Y_{C}$
Min.	0.00070	0.01075	0.00000
1stQ	0.04237	0.46139	0.00994
Median	0.10081	0.61189	0.03628
Mean	0.14544	0.60412	0.09800
3rdQ	0.20982	0.76187	0.12192
Max.	0.84055	1.00000	0.97111
Over 0.50	245	6944	336
Over 0.75	10	2698	62
Over 0.90	0	692	12

**Table 3 ijms-24-01301-t003:** Summary of similarity values with a $C_{m}$ over 0.75.

	$O_{v}$	$C_{m}$	$Y_{C}$
Min.	0.0034	0.7500	0.0005
1stQ	0.0886	0.7923	0.0336
Median	0.1891	0.8395	0.1049
Mean	0.2289	0.8501	0.1863
3rdQ	0.3395	0.9022	0.2737
Max.	0.8405	1.0000	0.9711

**Table 4 ijms-24-01301-t004:** Correlation between similarity indices and with genetic distance.

	$C_{m}$	$O_{v}$	$Y_{C}$	$D_{A}$
$C_{m}$	1.0000	0.4616	0.4256	−0.3442
$O_{v}$	0.4616	1.0000	0.9372	−0.7961
$Y_{C}$	0.4256	0.9372	1.0000	−0.8011
$D_{A}$	−0.3442	−0.7961	−0.8011	1.0000

**Table 5 ijms-24-01301-t005:** Summary of selected examples. Idx index of the simulated quasispecies pair, nHpl number of total haplotypes in the pair, nCm number of haplotypes in common, $C_{m}$, $O_{v}$, and $Y_{C}$ similarity indices, $D_{A}$ genetic distance.

Idx	nHpl	nCm	$C_{m}$	$O_{v}$	$Y_{C}$	$D_{A}$	Suppl. Figure
4213	16	8	0.9420	0.7477	0.8900	0.0180	S6
7426	15	9	0.9899	0.7232	0.8310	0.0711	S7
774	17	7	0.8611	0.0137	0.0071	0.7719	S8
5463	18	6	0.8521	0.0093	0.0039	0.6835	S9
3053	17	7	0.5741	0.2250	0.1789	0.4961	S10
5955	17	7	0.6312	0.1149	0.0417	0.5403	S11
1159	18	6	0.7983	0.5691	0.6454	0.1483	S12
2528	18	6	0.7946	0.6503	0.8052	0.0743	S13
345	16	8	0.7308	0.4823	0.6668	0.1199	S14

## Data Availability

The R code used to generate the simulated data, and used in the computations is provided in the Appendix A.

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
