# Peer review of "Quantifying In-Host Quasispecies Evolution"

_ijms, 2023, doi:10.3390/ijms24021301_

Round 1

Reviewer 1 Report

This manuscript aims to compare different similarity indices for characterizing quasispecies of viruses from sequence data. The authors come to the conclusion that all three examined indices capture different information, and that it is therefore most appropriate to use them in concert.

The aims and context of this manuscript could have been clearer. Virus evolution is not my main area of expertise, so this may have contributed. But in the methods it's stated that two quasispecies could have the same haplotype, which I initially found very confusing. If they have the same haplotype, then surely they must count as the same species? It was only after reading through the whole manuscript that it became clear to me that the main aim was to track changes in diversity over time, so that species A and species B represent samples from the same host at different time points. In such a case it's entirely possible that a shift in haplotype frequencies has occurred without necessarily entailing replacement of one quasispecies with another. I think this could be clarified.

Related to this main point, I think the practical applications of these methods could also have been explained in more detail. As an evolutionary biologist, I can see how tracking changes of different haplotypes over time in relation to different treatments could be valuable, since it lets us investigate whether a specific treatment is more effective against certain haplotypes. It would also allow investigation of whether the infection within a host seems to be evolving resistance against an ongoing treatment. None of this was brought up in the discussion, so I think including some broader perspectives of this type would be useful.

There are a number of small language errors throughout the manuscript. Here are the ones I caught on the first page:

Line 2: "specially" should be "especially"

Line 12: *an* RNA replication phase

Line 15: in the replication -> during replication

Line 18: loosing -> losing

Line 21: haplotypes distribution -> haplotype distribution

Line 25: entend -> extent

Line 33: in the paper -> throughout the paper

It would therefore be a good idea to get a professional copy editor to go through the language in the manuscript.

Author Response

Reviewer Comments:

Reviewer 1

This manuscript aims to compare different similarity indices for characterizing quasispecies of viruses from sequence data. The authors come to the conclusion that all three examined indices capture different information, and that it is therefore most appropriate to use them in concert.

The aims and context of this manuscript could have been clearer. Virus evolution is not my main area of expertise, so this may have contributed. But in the methods it's stated that two quasispecies could have the same haplotype, which I initially found very confusing. If they have the same haplotype, then surely they must count as the same species? It was only after reading through the whole manuscript that it became clear to me that the main aim was to track changes in diversity over time, so that species A and species B represent samples from the same host at different time points. In such a case it's entirely possible that a shift in haplotype frequencies has occurred without necessarily entailing replacement of one quasispecies with another. I think this could be clarified.

Answer: We sincerely thank rev #1 for the positive comments. In keeping with your suggestion, we have extended the explanations to clarify the points that were not clear and that will help broaden the putative audience of this work.

First of all, we have modified the title as “Quantifying in-host Quasispecies Evolution”, to avoid possible misunderstandings in the interpretation of this work. Also mentions to ‘in-host’ or ‘in a host’ have been introduced throughout the text (line 2, 7, 16, 28, 47, 165), to help in the understanding of what is being discussed. The abstract has been modified accordingly.

Lines 5 to 7: “The proposed methods treat the molecules in a quasispecies as individuals of competing species in an ecosystem, where the haplotypes are the competing species, and the ecosystem is the quasispecies in a host, …”

Lines 32 to 35: “With quasispecies simulated data we show their particularities and correlations, and use plots to help in the interpretation of results. Finally a clinical HEV dataset, from a recent publication, is used to illustrate the practical use of these indices.”

Lines 42 to 47: “To quantify the extend of changes in a quasispecies, we compare the quasispecies composition at two time points. The pairs of quasispecies used to illustrate the results and discussion are obtained by a simple simulation with a limited number of haplotypes whose frequencies vary randomly within given constraints, and where a random number of these haplotypes are common to both quasispecies. The simulation aims to obtain closely related quasispecies as we could find, a few weeks or months apart, in a host.”

Related to this main point, I think the practical applications of these methods could also have been explained in more detail. As an evolutionary biologist, I can see how tracking changes of different haplotypes over time in relation to different treatments could be valuable, since it lets us investigate whether a specific treatment is more effective against certain haplotypes. It would also allow investigation of whether the infection within a host seems to be evolving resistance against an ongoing treatment. None of this was brought up in the discussion, so I think including some broader perspectives of this type would be useful.

Answer: We would like to thank rev #1 for the comment. In keeping with your suggestion, we have introduced the application of these methods to a clinical case under a mutagenic treatment. We believe that this exercise will correct the deficiency pointed out by rev#1, also highlighted by rev#2 related to the possible applications of our work:

Lines 9-10: “The virtues of the proposed indices are finally shown on a clinical case.”

Lines 128 to 162: “2.3. A clinical case. This is the clinical follow-up of a patient chronically infected by HEV who underwent an off-label treatment with Ribavirin for three years [5]. The treatment involved three regimens (600, 800, and 1000mg/day) with discontinuations caused by adverse effects, followed by relapses.

This dataset is of particular interest here, because involves the follow-up of a patient infected by a zoonotic virus, HEV, treated with a mutagenic agent, with the follow-up spanning over three years of treatment. In this case, the naturally high genetic diversity of HEV quasispecies is enhanced by the treatment with a mutagen.

The behaviour of the three indices, in this case, is illustrated in Figure 10, where the similarities between each pair of sequential samples is shown, comparing sequential haplotype distributions on the left, and corresponding phenotype distributions on the right. The impact of the mutagenic treatment is evidenced by the sequential decrease in Cm, whose behaviour is smoother than that of Ov or YC. The continued decrease in Cm value indicates that the proportion of molecules with sequences corresponding to haplotypes common to the two compared quasispecies is shrinking, consistent with the expected results of a mutagenic treatment, which generates new variants at an enhanced rate. The new variants will increase in abundance or fade according to their replicative fitness. The drop in Cm is

especially marked when each treatment is initiated, especially those at 800 and 1000mg/day, but these are followed by a small correction upwards. On the other hand, despite the radical changes observed in the haplotype composition, the analysis by phenotypes composition shows that the functionality was maintained over a significant period of time, thanks to the generation of a rich set synonymous variants, and until the 800mg/day regimen took effect. The changes observed in phenotype composition near the end of treatment, together with the observed increase in viral load may indicate that, either some resistance to the treatment was generated, or that the rich set of synonymous haplotypes generated and selected during the treatments contributed to generate a more resilient quasispecies [6]. The similarity in the phenotype distribution between the end-of-treatment sample and that taken one year after results very high. This figure shows also that the indices Ov and YC are highly correlated, as previously shown with the simulated data. The Cm and Ov similarities in this dataset are plotted in Figure 11 over the 2D-density of the simulated data to show the correspondence between this clinical case and the simulated data.

The QFF profile of haplotypes and phenotypes, of this case, was presented and analysed in the previous publication [5], and provides an interesting complementary and consistent view of this quasispecies evolution.”

Lines 227 to 233: “The clinical case presented has given the opportunity to show a practical application of the proposed methods. This dataset with thousands of haplotypes in each sample, and coverages in the range of 5. x 104 to 5. x105 reads, shows a correlation between the three indices consistent with what has been observed with the more modest simulated pairs of quasispecies entailing very few haplotypes; nevertheless a critical aspect in the simulations was to ensure a close relationship between pairs of quasispecies, as it is the case in the follow-up of a patient, the main objective of this work.

Lines 239 to 243: “In the case of mutagenic treatments we recommend this method, combined with the method of quasispecies fitness fractions (QFF), and the Hill numbers profile (HNP) [5]. When the quasispecies evolution rate is low compared to mutagenic scenarios, the QFF may result insufficient to evidence changes in the quasispecies, and the proposed indices could result more sensitive to changes.”

Lines 293 to 306: “4.1.3. A clinical HEV case This dataset is taken from a recent publication [5], which shows the negative effects of early treatment discontinuation by a mutagenic agent of an HEV chronically infected patient. This dataset is used to show an example of application of the proposed method to a practical case. Briefly, this is the clinical follow-up case of a 27-year-old patient who acquired chronic HEV infection after undergoing two kidney transplantations. The patient received three different RBV regimens (600mg/day, 800mg/day, and 1000mg/day) with discontinuations caused by adverse effects, followed by relapses.

A single amplicon covering genomic positions 6323 to 6734 on the HEV ORF2 region was sequenced, for each of 13 sequential samples taken from May 2018 to June 2021. The coverage range of the final dataset is 53,307–503,770 reads, with a median of 328,271 reads per sample/amplicon, covering the full amplicon, and enabling the obtainment of amplicon haplotypes and corresponding frequencies. The number of haplotypes per sample are in the range 1,688-7,881, with a median number of 5,602.

There are a number of small language errors throughout the manuscript. Here are the ones I caught on the first page:

Answer: Thank you again for the effort and following your requirement we have deeply corrected the whole manuscript with the help of a native English-speaking professional.

Line 2: "specially" should be "especially"

Answer: Corrected (see line 2 in the marked version)

Line 12: *an* RNA replication phase

Answer: Corrected (see line 14 in the marked version)

Line 15: in the replication -> during replication

Answer: Corrected (see line 17 in the marked version)

Line 18: loosing -> losing

Answer: Corrected (see line 21 in the marked version)

Line 21: haplotypes distribution -> haplotype distribution

Answer: Corrected (see line 24 in the marked version)

Line 25: entend -> extent

Answer: Corrected (see line 28 in the marked version)

Line 33: in the paper -> throughout the paper

Answer: Corrected (see line 39 in the marked version)

It would therefore be a good idea to get a professional copy editor to go through the language in the manuscript.

Finally, we mention that the addition of the clinical case, which comes to answer suggestions by both reviewers, and improves notably the paper, represented the addition of some text and figures. In order to limit the burden of figures in the main text, the previous figures 10 to 13 have been moved to Supplementary Material as Supplementary Figures.

Reviewer 2 Report

This paper describes three similarity metrics (Cm, Ov, Yc) and one distance metric (Da) to quantify and visualize differences between RNA virus quasispecies. The methods are applied to simulated quasispecies as sampled from a distribution and as simulated to evolve following treatment. The description of the suggested methods is clear and the examples provided are illuminating to the differences and similarities of the different measures. Therefore this paper is of high interest to the viral research community as it provides means to describe quasispecies evolution. I suggest some minor points that could improve the paper and could possibly make it more relevant to a larger audience:

1) The distribution from which the quasispecies are simulated is likely to recapitulate characteristics of some viral quasispecies but not others. It would be a great addition if the authors could discuss which RNA virus quasispecies display similar metric values to the simulated ones. It would be nice to add points to the the graphs as in supplementary figures 4 and 5, not only of selected examples but of real datasets. 

2) In particular, HIV, which is an RNA virus of high interest is likely to present much higher diversity than in the simulated quasispecies presented in this study. It is likely that the proposed metrics would be of limited use in the case of HIV because of the substitution rate is so high that even in the same time point the major haplotype could have a relatively low frequency. Addressing the proposed metrics in the context of HIV would be highly desirable.

3) Minor text and spelling mistakes:
* Row 32:  In the context of NGS, we now each distinct...  The word now should probably be changed to 'denote'.
* Row 100: in the form of an shrinking...  Should be 'a'
*Row 136: between the tree indices... Should be 'three'

Author Response

Reviewer Comments:

Reviewer 2

This paper describes three similarity metrics (Cm, Ov, Yc) and one distance metric (Da) to quantify and visualize differences between RNA virus quasispecies. The methods are applied to simulated quasispecies as sampled from a distribution and as simulated to evolve following treatment. The description of the suggested methods is clear and the examples provided are illuminating to the differences and similarities of the different measures. Therefore this paper is of high interest to the viral research community as it provides means to describe quasispecies evolution. I suggest some minor points that could improve the paper and could possibly make it more relevant to a larger audience:

Answer: We sincerely thank rev #2 for the nice and positive comment and the suggestions that we have followed point-by-point and that have significantly helped to improve the paper.

1) The distribution from which the quasispecies are simulated is likely to recapitulate characteristics of some viral quasispecies but not others. It would be a great addition if the authors could discuss which RNA virus quasispecies display similar metric values to the simulated ones. It would be nice to add points to the graphs as in supplementary figures 4 and 5, not only of selected examples but of real datasets.

Answer: We agree with rev#2 concerning on whether the simulated data is close enough to the genetic variability observed in viruses like HIV, to be able to represent the inherent variability of RNA viruses. To put in clinical context the methods and the simulated data we added the data of a clinical follow-up of an HEV chronically infected patient treated with a mutagen, spanning three years of observation, and different treatment regimens. This is the clinical case of a RNA zoonotic virus whose high natural genetic diversity is enhanced by the action of the mutagen, and which could result in higher variability than expected of HIV. We hope this addition could satisfy the expressed concern also pointed out by rev#1.

Lines 128 to 162: “2.3. A clinical case. This is the clinical follow-up of a patient chronically infected by HEV who underwent an off-label treatment with Ribavirin for three years [5]. The treatment involved three regimens (600, 800, and 1000mg/day) with discontinuations caused by adverse effects, followed by relapses.

This dataset is of particular interest here, because involves the follow-up of a patient infected by a zoonotic virus, HEV, treated with a mutagenic agent, with the follow-up spanning over three years of treatment. In this case, the naturally high genetic diversity of HEV quasispecies is enhanced by the treatment with a mutagen.

The behaviour of the three indices, in this case, is illustrated in Figure 10, where the similarities between each pair of sequential samples is shown, comparing sequential haplotype distributions on the left, and corresponding phenotype distributions on the right. The impact of the mutagenic treatment is evidenced by the sequential decrease in Cm, whose behaviour is smoother than that of Ov or YC. The continued decrease in Cm value indicates that the proportion of molecules with sequences corresponding to haplotypes common to the two compared quasispecies is shrinking, consistent with the expected results of a mutagenic treatment, which generates new variants at an enhanced rate. The new variants will increase in abundance or fade according to their replicative fitness. The drop in Cm is

especially marked when each treatment is initiated, especially those at 800 and 1000mg/day, but these are followed by a small correction upwards. On the other hand, despite the radical changes observed in the haplotype composition, the analysis by phenotypes composition shows that the functionality was maintained over a significant period of time, thanks to the generation of a rich set synonymous variants, and until the 800mg/day regimen took effect. The changes observed in phenotype composition near the end of treatment, together with the observed increase in viral load may indicate that, either some resistance to the treatment was generated, or that the rich set of synonymous haplotypes generated and selected during the treatments contributed to generate a more resilient quasispecies [6]. The similarity in the phenotype distribution between the end-of-treatment sample and that taken one year after results very high. This figure shows also that the indices Ov and YC are highly correlated, as previously shown with the simulated data. The Cm and Ov similarities in this dataset are plotted in Figure 11 over the 2D-density of the simulated data to show the correspondence between this clinical case and the simulated data.

The QFF profile of haplotypes and phenotypes, of this case, was presented and analysed in the previous publication [5], and provides an interesting complementary and consistent view of this quasispecies evolution.”

Lines 227 to 233: “The clinical case presented has given the opportunity to show a practical application of the proposed methods. This dataset with thousands of haplotypes in each sample, and coverages in the range of 5. x 104 to 5. x105 reads, shows a correlation between the three indices consistent with what has been observed with the more modest simulated pairs of quasispecies entailing very few haplotypes; nevertheless a critical aspect in the simulations was to ensure a close relationship between pairs of quasispecies, as it is the case in the follow-up of a patient, the main objective of this work.”

Lines 239 to 243: “In the case of mutagenic treatments we recommend this method, combined with the method of quasispecies fitness fractions (QFF), and the Hill numbers profile (HNP) [5]. When the quasispecies evolution rate is low compared to mutagenic scenarios, the QFF may result insufficient to evidence changes in the quasispecies, and the proposed indices could result more sensitive to changes.”

Lines 293 to 306: “4.1.3. A clinical HEV case This dataset is taken from a recent publication [5], which shows the negative effects of early treatment discontinuation by a mutagenic agent of an HEV chronically infected patient. This dataset is used to show an example of application of the proposed method to a practical case. Briefly, this is the clinical follow-up case of a 27-year-old patient who acquired chronic HEV infection after undergoing two kidney transplantations. The patient received three different RBV regimens (600mg/day, 800mg/day, and 1000mg/day) with discontinuations caused by adverse effects, followed by relapses.

A single amplicon covering genomic positions 6323 to 6734 on the HEV ORF2 region was sequenced, for each of 13 sequential samples taken from May 2018 to June 2021. The coverage range of the final dataset is 53,307–503,770 reads, with a median of 328,271 reads per sample/amplicon, covering the full amplicon, and enabling the obtainment of amplicon haplotypes and corresponding frequencies. The number of haplotypes per sample are in the range 1,688-7,881, with a median number of 5,602.”

Moreover, the similarities between quasispecies in this clinical case have been plotted on the 2D-density plots of the simulated data, new Figure 11, and they match reasonably well.

2) In particular, HIV, which is an RNA virus of high interest is likely to present much higher diversity than in the simulated quasispecies presented in this study. It is likely that the proposed metrics would be of limited use in the case of HIV because of the substitution rate is so high that even in the same time point the major haplotype could have a relatively low frequency. Addressing the proposed metrics in the context of HIV would be highly desirable.

Answer: Thank you very much for the comment. Actually, the simulated data was not generated according to parameters observed in any virus, the generation aimed only at producing closely related quasispecies, similar to what could be observed in the follow-up of a single patient, with enough simplicity to be tabulated and plotted. However, HEV is an RNA virus having very mutation rates, similar or even higher than HCV or HIV. We have added a sentence in discussion to highlight this similarity between these viruses having a very high clinical impact.

Lines 183 to 192: “Here we have used simulated data aimed only at producing closely related quasispecies, similar to what could be observed in the follow-up of a single patient, with enough simplicity to be tabulated and plotted. However, to put in clinical context the methods here described, we have added the data of a clinical follow-up of an HEV chronically infected patient treated with a mutagen, spanning three years of observation, and different treatment regimens. Since HEV is an RNA virus having very high mutation rates on the range of 10-3 to 10-4 substitutions/base /replication cycle [7] similar to other highly clinical relevant viruses such as HCV or HIV, the tools here showed can be extrapolated to the vast majority if not all RNA viral infections.”

We agree that the major haplotype could be no so predominant over the other haplotypes. The simulated treatment, Figure 8, shows this case. We have removed ‘highly’ in line 337.

3) Minor text and spelling mistakes:
* Row 32: In the context of NGS, we
now each distinct... The word now should probably be changed to 'denote'.

Answer: Corrected (see line 38 in the marked version)

* Row 100: in the form of an shrinking... Should be 'a'

Answer: Corrected (see line 112 in the marked version)

*Row 136: between the tree indices… Should be 'three'

Answer: Corrected (see line 195 in the marked version)

Finally, we mention that the addition of the clinical case, which comes to answer suggestions by both reviewers, and improves notably the paper, represented the addition of some text and figures. In order to limit the burden of figures in the main text, the previous figures 10 to 13 have been moved to Supplementary Material as Supplementary Figures.